# Effects of Genistein on Common Kidney Diseases

**DOI:** 10.3390/nu14183768

**Published:** 2022-09-13

**Authors:** Qianwen Peng, Yuanyuan Li, Jia Shang, Haitao Huang, Yiming Zhang, Yueming Ding, Yipei Liang, Zhenxing Xie, Chaoran Chen

**Affiliations:** 1Institute of Nursing and Health, College of Nursing and Health, Henan University, Jinming Avenue, Kaifeng 475004, China; 2Department of Art, School of Kaifeng Culture and Tourism, Kaifeng 475004, China; 3School of Business, Henan University, Kaifeng 475004, China; 4School of Basic Medical Sciences, Henan University, Jinming Avenue, Kaifeng 475004, China

**Keywords:** genistein, kidney, diseases, review

## Abstract

Genistein is a naturally occurring phytoestrogen (soy or soybean products) that is classified as an isoflavone, and its structure is similar to that of endogenous estrogens; therefore, genistein can exert an estrogen-like effect via estrogen receptors. Additionally, genistein is a tyrosine kinase inhibitor, which enables it to block abnormal cell growth and proliferation signals through the inhibition of tyrosine kinase. Genistein is also an angiogenesis inhibitor and an antioxidant. Genistein has effects on kidney cells, some of the kidney’s physiological functions, and a variety of kidney diseases. First, genistein exerts a protective effect on normal cells by reducing the inflammatory response, inhibiting apoptosis, inhibiting oxidative stress, inhibiting remodeling, etc., but after cell injury, the protective effect of genistein decreases or even has the opposite effect. Second, genistein can regulate renin intake to maintain blood pressure balance, regulate calcium uptake to regulate Ca^2+^ and Pi balances, and reduce vasodilation to promote diuresis. Third, genistein has beneficial effects on a variety of kidney diseases (including acute kidney disease, kidney cancer, and different chronic kidney diseases), such as reducing symptoms, delaying disease progression, and improving prognosis. Therefore, this paper reviews animal and human studies on the protective effects of genistein on the kidney in vivo and in vitro to provide a reference for clinical research in the future.

## 1. Introduction 

Genistein, as a functional component existing in food, has received the attention of researchers for attenuating many diseases. Particularly in recent years, its beneficial effects on kidney diseases have constantly been proved in extensive research.

Genistein (4,5,7-Trihydrox-yisoflavone and 5,7-Dihydroxy-3-(4-hydroxyphenyl) chromen-4-one) is a naturally occurring compound that has an isoflavone compound structure. In 1899, genistein was first isolated from dyer’s broom. Its structure was established in 1926. It was chemically synthesized in 1928 [1]. Genistein is extensively distributed in edible plants and medicinal plants, such as a variety of legumes [2,3], Flemingia vestita, and F macrophylla [4,5,6]. It is also found in cell cultures of Maackia amurensis [7]. Additionally, it is a primary secondary metabolite in Trifolium species and Glycine max L [8]. Generally speaking, there are several purification methods for genistein, including maceration extraction, Soxhlet extraction, ultrasound-assisted extraction (UAE), microwave-assisted extraction (MAE), supercritical fluid extraction (SFE), and accelerated solvent extraction (ASE). The content of genistein in various foods is shown in Table 1.

It is well known that genistein is absorbed from plants in the form of glycoside, which is hydrolyzed to genistein though the intestinal microbiota or by phlorizin hydrolase (a small intestine brush-border lactase) [9,10]. Genistein is then absorbed by intestinal microflora in the form of aglycone; its oral bioavailability is approximately 10% [11]. The major signaling pathways involved in genistein metabolism are sulfation and glucuronidation. In addition, metabolites such as O-demethyldaidzein, dihydrogenistein, dihydrodaidzein, dihydroequol, and 6-hydroxy-O-demethylflavin generated by the liver and intestines have also been found in blood and urine [12]. Additionally, the kidneys, lungs, and heart can also effectively metabolize genistein [13]. After metabolism, genistein and its metabolites are completely excreted through the urethra and intestines within 24 h.

Genistein has many biological functions. First, genistein can exert an estrogen-like effect. This is due to the structural similarities between genistein and estrogen, making genistein an estrogen mimic in the body [14]. It mainly acts on estrogen receptors (ERs). However, unlike estrogen, genistein has more potential to act on ER β than ER α. Genistein is also used in treating postmenopausal symptoms, such as reductions in bone mass, vaginitis, and hot flashes [15], based on this effect. Second, genistein is a tyrosine kinase inhibitor. Therefore, genistein can be used to inhibit the development of many types of cancers, including breast cancer [16,17,18], brain cancer [19], colon cancer [17], prostate cancer [16,20], and cervical cancer [21], by blocking cell growth and proliferation signals mediated by tyrosine kinase. In addition, genistein is an angiogenesis inhibitor, as well as an antioxidant, etc. This means that genistein has other therapeutic effects besides those on postmenopausal symptoms and cancers, including on cardiovascular disease [22], obesity [23], and diabetes mellitus [24], as well as antidepressant and anxiolytic effects [25]. Recently, an increasing number of studies have shown that genistein has a variety of protective effects on kidney cells and kidney diseases [26].

**Table 1 nutrients-14-03768-t001:** The content of genistein in various foods.

Foods	Content (μg/100 g Hydrated Portion)	Reference
Soybean	26,800–102,500	[27]
Kidney bean	18.0–518.0	[27]
Chickpea	69.0–214.0	[27]
Pea	0–49.7	[27]
Lentil	7.0–19.0	[27]
Kudzu leaf	2520	[27]
Kudzu root	12600	[27]
Black gram	1900	[28]
Alfalfa	5.0	[28]
Peanut	8.0	[29]
Caraway seed	64.0	[28]
Sunflower seed	13.9	[30]
Barley	7.7	[28]
Broccoli	8.0	[28]
Cauliflower	9.0	[29]

The kidney is one of the most important organs: it is involved in the homeostasis of the whole body, such as regulating the acid–base balance, electrolyte concentrations, osmotic pressure, and blood pressure, eliminating toxins, etc. Additionally, the kidneys can also produce and secrete a variety of endocrine hormones, including renin, erythropoietin, and calcitriol. There are different kinds of kidney diseases treated in the clinic. When kidney diseases occur, they can cause serious harm to the body’s functioning and even lead to death. In the clinical setting, there are many methods for treating kidney diseases, but the vast majority of kidney diseases are irreversible and difficult to cure, even with high-cost treatment. Therefore, it is extremely necessary to develop new strategies to prevent the occurrence and development of kidney diseases. Therefore, it is urgent to summarize these data (human and animal studies) to inspire future research and possible clinical applications of genistein to attenuate kidney diseases.

## 2. The Role of Genistein on Pathologies of Kidney Cells

### 2.1. The Effects of Genistein on Mesangial Cells

Glomerular mesangial cells (MCs) lie between and around the glomerular capillaries, creating a support structure for the tuft of capillaries, and are involved in the production of inflammatory mediators, such as cytokines, macromolecules, and immune complexes. Additionally, MCs have a contractile function, and under the influence of vasoconstrictors and vasodilators, they affect filtration by changing the surface area of the filtration slits through contraction and relaxation [31]. In the presence of pathological conditions (e.g., growth factors, inflammation, glomerular capillary hypertension, and glucose toxicity), the mesangial cells are activated, resulting in a glomerular pathology [32].

Previous studies have shown that genistein can inhibit the abnormal activation of mesangial cells in vitro, potentially preventing the occurrence of kidney injuries [26]. Inhibition of inflammation or proliferation by impeding the production and effect of proliferation stimulators (e.g., EGF and PDGF) and inflammatory factors (e.g., TGF-β, IL-1β, and PGE2) is considered to be involved in the effect of genistein, as summarized in a previous review [26]. However, the related mechanisms were not further discussed, except for the ability of genistein to inhibit tyrosine kinase. In recent years, a large number of studies have further expanded the understanding of the effects of genistein on MCs, and the relevant mechanisms have also been deeply explored (Figure 1). First, genistein is able to inhibit inflammation in MCs via the suppression of interleukin-1 beta (IL-1β)/MCP-1, NF-κB (nuclear factor-kappaB)/matrix metalloproteinase-9, and Gro chemokine transcription, and increasing group II phospholipase A2 transcription (PLA2) [33,34,35,36] prevented the activation of the glutamine:fructose-6-phosphate amidotransferase (GFAT) promoter and disrupted AGE-RAGE binding [37,38]. Second, genistein can inhibit the abnormal proliferation of MCs through multiple pathways, including by inhibiting the synthesis of DNA [39,40] and calcium (Ca^2+^) accumulation, which impedes proliferation-stimulating receptors (e.g., 5-HT2A receptors and erythropoietin) [41,42] or the formation of complexes with other components (e.g., PLC-γ1 and PDGF-β receptor membrane complex; growth factor receptor-binding protein 2, son of sevenless, and PDGF-β receptor complex) [43,44,45], preventing the internalization of the angiotensin II receptor and its downstream responses (e.g., PAI-1 mRNA; PLC-gamma 1/IP3 and Ca^2+^) [46,47,48,49]; attenuating vascular permeability factor (VPF)/cGMP and reducing MAPK signals through the inhibition of ERK2/Elk/(AP-1 and Fos) [50,51], PTK/PKC-Ras--MAP kinase activity, and autophosphorylation of pp60c-src [52,53,54,55]; and preventing the secretion and expression of TGF-β1 [56,57]. Third, genistein can restrain the pathological apoptosis of MCs by reducing the activation of c-Jun phosphorylation (SAPK) [58,59] by angiotensin II/(ATP and UTP) and nitric oxide. The production of nitric oxide can also be suppressed by genistein [60,61]. Moreover, genistein can inhibit remodeling, which results from inhibited collagen gel contraction, reconstruction, and degradation by impeding ERK and collagenase mRNA [62,63]. For collagen synthesis, low-dose genistein mimics the effect of estrogen through estrogen receptors to inhibit the synthesis of collagen in mesangial cells rather than by inhibiting tyrosine kinase, which is effective for inhibiting remodeling [64]. High-dose genistein promotes collagen synthesis mainly by inhibiting tyrosine kinase and collagen degradation, which is detrimental to inhibiting remodeling [65]. Therefore, the dose relationship should be considered when genistein is used to remodel MCs. Therefore, genistein can block the development of kidney diseases by improving the function of mesangial cells.

### 2.2. The Effects of Genistein on Endothelial Cells

As is well known, the urinary function of the kidney is accomplished by the nephron and the collecting duct. Nephrons are tiny or microscopic structural and functional units of the kidneys. They consist of a kidney corpuscle and a kidney tubule. The kidney corpuscle consists of a cluster of capillaries called a glomerulus and a cup-shaped structure called Bowman’s capsule. The kidney tubules extend from the capsule. The capsule is connected to the tubule and is composed of epithelial cells with a lumen. The function of the kidney corpuscle is to filter the original urine, while the kidney tubule is responsible for processing and removing the filtered fluid. The collecting duct system consists of a series of tubules and ducts that connect the nephron to the calyx or directly to the kidney pelvis. Collecting tubes participate in the balance of electrolytes and liquids through reabsorption and excretion. Studies have shown that genistein plays a role in nephrons and collecting ducts, including glomeruli, proximal tubule cells, distal tubule cells, and collecting duct cells (Figure 2).

In glomerular endothelial cells, genistein can inhibit inflammation caused by GEC MCP-1 mRNA expression induced by LysoPC [66]. In the proximal tubule, genistein plays a more diverse role. Genistein can not only protect dopamine precursors, reduce insulin uptake, and increase insulin accumulation in proximal tubular cells but also protect dopamine receptors, prevent their serine phosphorylation, and inhibit their mediated NKA activity [67,68,69,70], thereby maintaining kidney function and reducing the occurrence of disease. Additionally, genistein can inhibit the Na+-glucose cotransporter (e.g., IL-6) and induce the Na+-glucose cotransporter (e.g., Ang II and EGF); the former may be due to the inhibition of abnormal proliferation induced by inflammation [71,72,73]. Other mechanisms include suppressing the Na+/H+ exchanger (NHE, including NHE1/3), which can balance proliferation, differentiation, and cell forms [74,75,76]; restrain inflammation (↓MCP-1) and downstream signals (↓HIF-1) [77,78]; block DNA synthesis (e.g., thrombin and HGF) and DNA damage (hypoxia or chemical) [77,79]; prevent PAI-1 (induced by Ang II and 15d-PGJ2) production to reduce the emergence of fibrosis [80,81]; inhibit HSP70 after heat shock and prevent the disorder of normal protein formation caused by its excess [82]; inhibit protein hyperphosphorylation (β-catenin and plakoglobin phosphorylation) to maintain tight junctions [83]; restrain PAH, L-alanine transport (↑EGF), H(+)-ATPase (Ang II), and Pi, acting as a tyrosinase inhibitor, indicating that they all need tyrosinase activation (but with regard to PI uptake, genistein blocks the inhibition of EGF on its uptake, because EGF inhibits Pi uptake by activating EGF receptors) [84,85,86,87,88]; block the decrease in Mg^2+^; and prevent the production of nitric oxide [89]. In distal tubules, genistein mainly regulates ions and maintains the acid–base balance, e.g., by decreasing Na+ transport (e.g., hyposmolality, insulin, and aldosterone [90,91,92]) and HCO3- (NGF and hyposmolality [93,94]). In collecting ducts, genistein can inhibit the excessive activity of H-K-ATPase induced by isoproterenol and maintain the balance of H+-K+ exchange [92], impeding apoptosis in hypertonic conditions (NaCl and urea) [95], suppressing the expression of COX-2 to maintain the stability of blood volume [96], and increasing the activity of SMIT (sodium/myo-inositol cotransporter) so as to increase the activity of cells [97]. Additionally, genistein can inhibit IGF-I-induced (insulin-like growth factor-I) nitric oxide in endothelial cells of interlobular vessels [98]. However, the effects of genistein on kidney tubules and collecting ducts are not all favorable. For example, in proximal tubule cells, genistein blocks the stimulating effect of Ang II on the expression of the Pax-2 gene, which plays an important role in kidney repair (lack of expression leads to apoptosis in cells [99]). In the distal tubule, genistein reduces the synthesis of sulfoglycolipid, which can protect cells from a hypertonic environment [100]. This indicates that it is necessary to further study the effects of genistein on kidney tubules and collecting ducts.

### 2.3. The Effects of Genistein on Podocytes

Podocytes are cells that wrap around glomerular capillaries and are located in the kidney’s Bowman’s capsule, forming the epithelium of Bowman’s capsule, which is the third layer of blood filtration [101]. Studies have shown that the decrease in or depletion of podocytes and their morphological changes play an important role in the development of progressive nephropathies, such as glomerulosclerosis and diabetic nephropathy [102,103,104]. A number of studies have shown that genistein has multiple favorable and unfavorable effects on podocytes (Table 2). The beneficial effects include the inhibition of inflammation (EVs and IL-1β) and the promotion of autophagy. The mechanism is the reduced activation of the inflammasome via the inhibition of TGβ1/FAK in normal podocytes and the induction of autophagy through the inactivation of mTOR in HG (high-glucose)-treated podocytes. The unfavorable effect is the promotion of the loss of podocytes under the effect of fluid shear force and the apoptosis of podocytes in the presence of integrinα3β1, the reorganization of the actin cytoskeleton, and circular ruffles. The mechanism of the latter effect inhibits FAK/integrinα3β1–ECM interaction. The double-sided effect of genistein on podocytes indicates that genistein should be used with caution; we can increase the use of genistein in terms of its known beneficial effects and avoid unfavorable factors so as not to cause greater damage to the kidneys.

## 3. The Effects of Genistein on Kidney Physiology (Figure 3)

### 3.1. The Effects of Genistein on Renin

Renin, which is secreted by granule cells of the paracellular apparatus in the kidney, participates in the activation of the renin–angiotensin–aldosterone system and regulates the mean arterial blood pressure. Studies have shown that genistein may be able to maintain the balance of blood pressure by affecting the secretion of renin. In a previous study [110], researchers found that genistein treatment suppressed the IL-1β-mediated attenuation of renin gene transcription in vitro. In another in vitro study on the effect of epidermal growth factor (EGF) on the secretion of renin, the researchers observed that genistein treatment eliminated the inhibitory effect of EGF on renin secretion [111]. Therefore, genistein may be used to maintain normal blood pressure by restoring renin release, which is suppressed in some conditions, such as inflammation (the former study), and endogenous vasoconstrictors (e.g., EGF). However, more studies are needed to explore the mechanism.

### 3.2. The Effects of Genistein on Regulating Calcium and Phosphate

Ca^2+^ and phosphate (Pi) are two key factors in maintaining the balance of bone formation and bone loss as well as other physiological functions (muscle contraction, blood coagulation, and neurotransmitter release [112]). When the balance of calcium is disrupted, bone pathologies or other dysfunctions in the body will occur. Two studies explored the protective effect and potential mechanism of genistein on the dynamic balance of calcium in male animals. A study showed that genistein treatment (30 mg/kg b.m/day) decreased Ca^2+^ content in urine and increased 25 (OH) vitamin D content in serum. Additionally, increased expression of the Klotho gene and protein also contributes to calcium reabsorption, possibly activating the TRPV5 Ca^2+^ channel in the kidneys of orchidectomized rats (an andropause model) [113]. Additionally, the downregulated expression of FGFR and PTH1R reduces Pi reabsorption via the inhibition of the activity of the NaPi 2a cotransporter. However, in another study, genistein treatment (10^−6^ M) eliminated the effect of testosterone-enhancing kidney Ca^2+^ reabsorption in the distal luminal membrane of rabbit kidneys [114]. Therefore, the dual effects of genistein in regulating Ca^2+^ and Pi balances depend on the endocrine hormone (e.g., testosterone). Thus, genistein may be used carefully in regulating Ca^2+^ under different conditions.

### 3.3. The Diuretic Effect of Genistein

Recent studies have shown that genistein has a diuretic effect by decreasing vasodilatation instead of decreasing the glomerular filtration rate. A study [115] showed that there is a diuretic effect of genistein (15 mg/kg) in rats, which is similar to that of furosemide. However, the effective dose was 3–5 times lower than that of furosemide (a diuretic). However, genistein had no significant effect on the glomerular filtration rate, despite decreasing kidney vascular resistance. While the diuretic mechanism of genistein has not been further explained, another study showed that the diuretic effect of genistein is due to the reduction in the membrane Na-K-Cl cotransporter concentration [116]. Accordingly, genistein may be used as a new diuretic in future treatment, and more in-depth clinical research is needed in the future.

### 3.4. The Effects of Genistein on Nephron Barrier

It is well known that the filtration function of the kidney depends on the normal kidney permeability barrier. However, when the permeability is pathologically damaged, pathological filtration dysfunctions occur, e.g., increased or decreased filtration. Studies have shown that genistein protects the permeability barrier of the kidney. In one study [117], genistein treatment (100 μM, 1 h) significantly inhibited the increase in Pa (albumin permeability) in acute glomerular inflammatory injury induced by SNAP (s-nitroso-N-acetyl-penicillamine) in rats in a dose-dependent manner. This effect may be achieved by inhibiting tyrosine kinases. However, another study found the opposite results that genistein (50 or 100 μM: 6 h) was not conducive to the repair (the relocalization of ZO-1 and occludin to the tight junction) of kidney tubular tight junction damage caused by oxidative stress and decreased ATP [118]. The differences in the effect of genistein may be due to the fact that genistein can protect the kidney from the acute kidney injury barrier when it is not seriously damaged, but it is not conducive to weakening the silent junction damage that has already occurred.

**Figure 3 nutrients-14-03768-f003:**
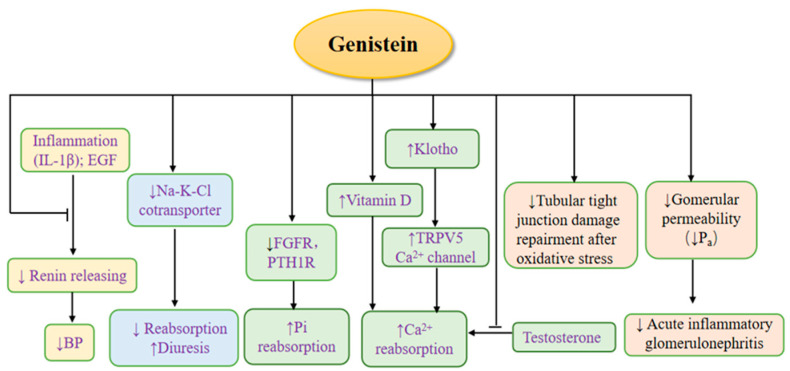
The effects of genistein on kidney physiology. IL-1β: interleukin-1 beta; BP: blood pressure; FGFR: fibroblast growth factor receptor; PTH1R: parathyroid hormone 1 receptor; TRPV5: transient receptor potential vanilloid 5; Pa: albumin permeability, ↓: inhibit; ↑: upregulate.

## 4. The Effects of Genistein on Common Kidney Diseases

### 4.1. The Effects of Genistein on Acute Kidney Injuries

#### 4.1.1. LPS

Lipopolysaccharide (LPS), which is a component of the outer membrane of Gram-negative bacteria, can stimulate the innate immune response. Additionally, LPS, as a special antigen [119], is able to induce endotoxemia [120], sepsis [121], and even DIC [122]. Therefore, LPS is usually used to induce different animal models, such as neuroinflammation [123], endometritis [124], memory impairment [125], and acute kidney injuries [126]. LPS is co-sensed by the CD14 protein and TLR4-MD2 complex at the plasma membrane [127]. Ligated TLR4 then activates the transcription of immune factors (e.g., cytokines and chemokines) mediated by NF-κB and IRF3 through MYD88 and TRIF. This subsequently upregulates the expression of various inflammatory mediators, such as tumor necrosis factor α (TNF-α), interleukin-6 (IL-6), and IL-1β.

Several studies have shown that genistein treatments can significantly improve acute kidney injuries caused by LPS in vitro and in vivo. For example, an animal experiment showed that genistein pretreatment (10 mg/kg) dramatically suppressed fractalkine expression through LPS-mediated TNF-α in the arterial endothelial cells of rat kidneys [128]. In addition, in another animal experiment [129], genistein (10, 30, and 90 mg/kg injected i.p. 0.5 h before the LPS injection and 2 h and 8 h after the LPS injection) significantly improved the morphological structure, fiber protein deposition, function indicators (inhibition of blood urea nitrogen, BUN), and inflammation reaction (↓NF-κB, IL-6, and IL-1) in kidneys.

Additionally, in a clinical study [130] of hemodialysis patients with end-stage kidney disease, genistein treatment (25 μM for 24 h) inhibited LPS-induced inflammation activation (downregulated TNF-α levels) in whole blood and monocytes in vitro. Therefore, genistein may be used to attenuate acute kidney injury induced by LPS. Further clinical experiments need to be carried out.

#### 4.1.2. The Effects of Genistein on Kidney Ischemia/Reperfusion Injury

Ischemia refers to the partial or complete occlusion of blood perfusion in tissues or organs. It can lead to a lack of oxygen and other nutrients as well as the accumulation of metabolic wastes. If perfusion is not restored in time, ischemia can quickly lead to tissue necrosis [131]. Early reperfusion is the preferred intervention to prevent pathological processes. However, reperfusion after ischemia often causes inflammation and apoptosis through oxidative stress [132]. Kidney I/R injury usually occurs due to shock [133], sepsis [134], low perfusion [135], and kidney transplantation [136]. Kidney I/R injury will cause direct damage to tubules and blood vessels. Then, glomeruli acute kidney injury will occur as a result of oxidative stress, inflammation, and mitochondrial dysfunction, leading to a sharp decrease in kidney function and even an increase in mortality [137]. As the kidneys maintain the water balance of the human body, its I/R process can also cause severe damage to other organs [138,139].

Several studies have shown that under normal circumstances, genistein can inhibit the contraction of kidney blood vessels, including glomerular arterioles, afferent arteries, or vascular beds [140,141,142]. In addition, genistein can inhibit norepinephrine-induced vasoconstriction, but this effect is not completely dependent on extracellular Ca^2+^ [143]. It may be partly related to the inhibition of the large-conductance K^+^ channel, according to another study [144]. Therefore, genistein may attenuate kidney ischemic injury by inhibiting the contraction of kidney blood vessels. For instance, in one study [145], genistein treatment (3 μ for 45 min) significantly inhibited kidney vasoconstriction induced by G1 (a G protein-coupled estrogen receptor1 agonist) in isolated perfused (perfusion immediately after separation) rats. The effect of genistein is exerted through the blocking of the estrogenic effect. However, in another study [146], genistein (10 mg/kg, 30 min before hemorrhagic shock) reduced kidney injury during ischemic shock in rats by inhibiting inflammation. However, the effect of genistein on vasoconstriction was not studied.

Encouragingly, some studies have shown that genistein pretreatments at a shorter time before I/R can attenuate kidney I/R injury in animal models. One study showed that genistein treatment (5, 10, or 15 mg/kg, 30 min before ischemia) significantly reduced kidney cell death through the stimulation of kidney cell proliferation mediated by the up-regulation of SIRT1 expression (exerting cytoprotective effects) in kidney I/R mice (45 min/24 h) [147]. In another study, genistein (15 mg/kg body weight, i.p.), administered 30 min before ischemia and 1 h after ischemia, reduced kidney injuries in kidney I/R (45 min/24 h) rats by reducing the inflammation response (decreasing TLR-4 and TNF-α expression) and oxidative stress (intensifying antioxidant ability) [148]. However, in a previous study [149] exploring the effects of EPO on kidney ischemic injury in ischemic shock mice, genistein treatment (10 mg/kg, 2 h before ischemia) alone increased kidney ischemic injury in model mice, and treatment could also reverse the beneficial effects of EPO in attenuating the kidney injury of model mice. The adverse effect in the latter study may be due to the time point of genistein administration; that is, genistein administration before ischemia may be the reason behind this. The exact mechanism still requires further investigation in the future.

### 4.2. The Effects of Genistein on Kidney Cancer Cells

Renal cancer affects nearly 300,000 people worldwide each year, causing more than 100,000 deaths annually [150]. However, owing to the lack of obvious symptoms in the early stage, only 30% of patients are diagnosed according to clinical symptoms in the advanced stage, including local symptoms (acute or chronic flank pain, gross hematuria, and palpable abdominal mass and varicocele) and paraneoplastic disorders (e.g., hematuria, high blood pressure, anemia, cachexia, weight loss, hypercalcemia, etc.) [151]. Increasing 5-year survival rates after diagnosis required the development of new drugs for systemic treatment. In recent years, research on the effect of genistein on renal cell carcinoma has mainly been focused on human kidney cancer cells. A few studies have conducted animal experiments. The main effects of genistein on kidney cell carcinoma are the decrease in the proliferation and migration of cancer cells, the inhibition of the neovascularization of solid tumors, the retardation of the growth of RCC cells, the induction of cell apoptosis, and the induction of tumor growth. These effects are achieved through the following mechanisms (Table 3): inhibiting cancer cell proliferation by ① increasing the expression of CDKN2a and CDKN2a methylation, ② suppressing EED (embryonic ectoderm development) levels in PRC2, ③ inhibiting HOTAIR (HOX transcript antisense RNA)/PRC2 (polycomb repressive complex 2) interaction, ④ suppressing HOTAIR/PRC2 recruitment to the ZO-1 promoter and enhanced ZO-1 transcription, and ⑤ inhibiting SNAIL transcription by reducing HOTAIR/SMARCB1 (subfamily B member 1) interaction and decreasing EGF; impeding angiogenesis by decreasing the expression of the angiogenic factors vascular endothelial growth factor (VEGF) and basic fibroblast growth factor (FGF); suppressing miR-1260b expression (in A-498 cells, inhibiting Wnt signaling), which is highly expressed in kidney cancers; reducing the basal activity of sulfotransferase; and inducing cell apoptosis.

### 4.3. The Effects of Genistein on Diabetic Nephropathy

Diabetic nephropathy is one of the most common and serious complications of diabetes, which is related to an increase in morbidity and mortality in patients with diabetes [162]. It is considered to be one of the leading causes of chronic kidney disease (CKD) and end-stage renal disease (ESRD) worldwide. Therefore, diabetic nephropathy has become a global health problem [163,164]. The etiology of diabetic kidney injury is complex and has multiple factors, including hyperglycemia, hypertension, dyslipidemia, the production of inflammatory cytokines, and oxidative stress [165]. Some studies have suggested that genistein may be a good protective agent against diabetic nephropathy in type 1 and 2 diabetes models (Table 4). The mechanism of genistein to improve diabetic nephropathy mainly includes the following aspects: inhibiting AGE (advanced glycation end product) formation by trapping MGO to form adducts and upregulating the expression of glyoxalase I and II and aldose reductase (AR); improving levels of fasting blood glucose (FBG) levels; reducing kidney inflammation (decreasing levels of interleukin-6, TNF-α, and C-reactive protein; attenuating kidney oxidative stress (nuclear-related factor E2, heme oxygenase-1/HO-1, glutathione peroxidase, and superoxide dismutase isoforms); inhibiting fibrosis-related markers (protein kinase C, protein kinase C-beta II, and transforming growth factor-beta I); and reducing apoptosis and normalizing vasoconstriction induced by agonists (norepinephrine, endothelin-1, and Ang II).

Additionally, a recent in vitro study [166] showed that genistein can block the process of decoy receptor 2 interacting with peroxiredoxin 1, which may be related to the amelioration of kidney fibrosis in diabetic nephropathy progression. Another study [167] found that genistein had an inhibitory effect on AR purified from sheep kidneys, which plays a vital role in the development of diabetic nephropathy. Genistein also had synergistic antioxidant effects, alongside other substances (e.g., resveratrol), on high-glucose-incubated Madin-Darby canine kidney (MDCK) epithelial cells [168]. The mechanism was the reduction in ROS via the inhibition of nicotinamide adenine dinucleotide phosphate (NADPH) oxidase expression and increased γ-glutamylcysteine synthetase expression. The genistein–chromium (III) complex can also improve pathological injuries in the kidneys of diabetic mice, which provides a new way to apply genistein treatment [169]. Therefore, the combined use of genistein may have better prospects.

In addition, excessive sugar intake can induce multiple metabolic syndromes, such as insulin resistance, hyperglycemia, hypertriglyceridemia, hypertension, and also kidney damage associated with oxidative stress. Genistein treatment (1 mg/kg/day, i.p., for 60 days) significantly improved insulin sensitivity, glomerular function (increased clearance rates), and kidney parenchymal injury, as confirmed histologically, in a rat model of kidney injury induced by a high-glucose diet [170]. Additionally, genistein treatment can also impede the occurrence of nephropathy through a high-fructose diet. However, genistein treatment (2, 6, and 20 mg/kg for 90–180 days) had little effect on the diabetic nephropathy of females, according to a previous study [171]. Although a genistein-containing diet could protect female mice from developing type 1 diabetes, only kidney weights were increased, and the histopathology of kidneys was not ameliorated. This may be because the female rats used in this study with a certain amount of estrogen in their bodies altered the effect of genistein.

**Table 4 nutrients-14-03768-t004:** The effects of genistein attenuating diabetic nephropathy.

Animal	Diabetes Models	Treatments (Genistein)	Effects and Mechanisms	Ref.
C57BL/6J mice	High-fat diet	0.25% genistein in diet for 18 weeks	Inhibiting AGE formation by trapping MGO to form adducts and upregulating the expression of glyoxalase I and II and aldose reductase in kidney to detoxify MGO	[172]
Albino rats	Alloxan-induceddiabetes	20 mg/kg/day for 30 d	Normalizing kidney function (biomarkers: creatinine and BUN) by downregulating inflammatory responses (↓IL-6, TNF-α, and C-reactive protein in serum)	[173]
Mice	Streptozotocin-induced diabetes	10 mg/kg, i.p. three times a week for 10 weeks	Reducing kidney inflammation, oxidative stress, and apoptosis	[174]
ICR mice	Alloxan-induced diabetes	0.25 and 1 mg/g in diet for 2 weeks	Improving levels of FBG and attenuated kidney oxidative stress; decreasing inflammatory and fibrosis-related markers	[175]
Wistar rats	STZ-induced diabetes	4 mg/kg b.w/day, i.p. for 7 d	Protecting against kidney dysfunction, lowering blood glucose levels, and protecting against kidney dysfunction	[176]
KKAy mouse	Type 2 diabetes	12 mg /kg, oral gavage, once a day for 3 months	Inhibiting inflammatory responses, repressing HGA-induced activator protein 1 activation and oxidase stress generation, and reducing NADPH oxidase (NOX) gene expression	[177]
Wistar rats	STZ-induced diabetes	1.5 mg/kg/alt diem for 4 weeks	Normalizing vasoconstriction induced by agonist (norepinephrine, endothelin-1, and Ang II)	[178]

### 4.4. The Effects of Genistein on Hypertensive Kidney Disease

At present, hypertension affects approximately 30% of the general population and causes damage to multiple organs [179]. It is worth noting that hypertension is a risk factor for the progression of kidney damage (hypertensive kidney disease) [180]. Hypertensive kidney disease is considered to be one of the consequences of long-term and poorly controlled hypertension. In turn, kidney disease can cause or aggravate hypertension. Hypertensive kidney disease is a leading cause of end-stage kidney failure, second only to diabetic kidney disease [179]. Many studies focus on how to reduce hypertension. Therefore, several studies have shown that genistein treatment is able to improve hypertension and kidney injuries in hypertensive animal models (Table 5), which is mainly through the reduction in blood pressure and kidney vascular tension. The mechanisms are as follows: lowering BP (restored ACE, PKC-βII and eNOS/NO, and cGMP); blunting the dose-related increase in arterial pressure; and reducing kidney vascular resistance relative to vehicle in isolated perfused kidneys. Therefore, genistein can protect the kidney by reducing kidney hypertension and maintaining the ultrastructural integrity of the kidney. It may provide a therapeutic option for the treatment of kidney hypertension in the future.

### 4.5. The Effects of Genistein on Kidney Injury by Medications and Irradiation (Table 6)

Medications are a relatively common cause of kidney injury [184,185,186]. At present, the epidemiology of drug-induced nephrotoxicity is mainly based on the AKI (acute kidney injury)-related literature. Drug-induced nephrotoxicity in adults was approximately 14–26%, while in the pediatric population, 16% of AKI hospitalizations were caused by drugs [185,186]. Common nephrotoxic drugs include cisplatin, cephaloridine, and gentamicin. Cephaloridine can produce dose-related nephrotoxicity when administered in high doses [187]. Its nephrotoxicity can be distinguished by the degree of acute proximal tubular necrosis in laboratory animals and humans [188]. A study showed that genistein (25 μg/mL, 2 h) inhibited increases in LDH leakage (an index of cellular injury) and lipid peroxidation in LLC-PK1 cells exposed to cephaloridine [189]. Gentamicin, an aminoglycoside antibiotic, is highly effective in the treatment of severe Gram-negative infections. However, it has nephrotoxic effects on the epithelial cells of the proximal tubules. Genistein treatment (10 mg/kg/day i.p.) produced reno-protective effects (decreased serum levels of Kim-1, cystatin C, LDH, and GGT) in gentamicin-induced acute kidney injury [190]. Cisplatin is one of the most effective and active cancer chemotherapy drugs for the treatment of various malignancies [191], inducing nephrotoxicity mainly through the reaction of platinum and thiol protein groups [192]. Genistein treatment (10 mg/kg/day orally for 3 days) inhibited oxidative stress (reduced reactive oxygen species production) and the inflammation response by decreasing the expression of intercellular adhesion molecule-1 (ICAM) and monocyte chemoattractant protein-1 (MCP-1) proteins and the translocation of the p65 subunit of NF-κB [193]. Genistein also decreased cisplatin-induced apoptosis (regulating p53 induction in the kidney). Therefore, genistein may protect the kidney from drug-induced nephrotoxicity.

Radiation therapy (RT) is one of the most common and important cancer treatment techniques [194]. As a radiosensitive organ, the kidney is inevitably exposed to radiation in the abdominal cancer treatment room. High-dose radiation can cause kidney damage (even radiation nephropathy), including increased vascular permeability, perfusion disorders, inflammation, and fibrosis [195,196]. In a previous study, genistein treatment 24 h before RT decreased the incidence of kidney tubular atrophy and the level of malondialdehyde (MDA) in mouse kidneys. Therefore, genistein supplementation prior to irradiation can be used to protect mice against radiation-induced nephrotoxicity.

**Table 6 nutrients-14-03768-t006:** The effects of genistein attenuating injury caused by medications and irradiation.

Animal	Model	Treatments (Genistein)	Effects and Mechanisms	Ref.
LLC-PK1	Cephaloridine-induced kidney injury	25 µg/mL preincubated for 2 h	Inhibiting increases in LDH leakage and lipid peroxidation in LLC-PK1 cells exposed to cephaloridine	[189]
Sprague-Dawley rats	p-Nonylphenol-induced polycystic kidneys	0.005 μM/10 μL for 35 d	Modulating the development of PKD induced by dietary NP in rats	[197]
Mice	Cisplatin-induced kidney injury and cisplatin-treated normal human kidney HK-2 cells	10 mg/kg orally once a day for 3 d	Decreasing oxidative stress (reactive oxygen species), inflammation (ICAM, MCP-1, and NF-κB), and apoptosis (regulating p53 induction)	[193]
Wistar albino rats	Gentamicin-induced acute kidney injury rats	10 mg/kg/day, i.p, one week before gentamicin treatment, for 17 d	Decreasing serum levels of Kim-1, cystatin C, LDH, and GGT	[190]
Swiss albino mice	A single dose of 6 Gy γ-radiation (Co60)	200 mg/kg, subcutaneous injection, for 24 weeks	Decreasing the incidence of kidney tubular atrophy and the level of MDA	[198]

### 4.6. The Effects of Genistein on Kidney Fibrosis

Kidney fibrosis, an inevitable consequence of chronic kidney disease [199], is caused by multiple diseases, such as diabetes, obstructive urinary tract, glomerulonephritis, glomerulosclerosis, kidney hypertrophy, and mutation [200]. In the process of kidney fibrosis, many events in kidney cells occur, including inflammation, oxidative stress, fibroblast activation and the expression of fibrosis-related cytokines, vascular remodeling and hypertension, kidney tubular apoptosis, and autophagy [201]. Ultimately, this leads to the loss of kidney function and the replacement of kidney parenchyma by scar tissue [202].

In order to reduce kidney fibrosis, it is important to find novel and reliable therapeutic methods. Several experiments have shown that genistein can improve kidney fibrosis in vivo and in vitro. In vivo, genistein treatment can significantly alleviate fibrosis in several animal models of kidney interstitial fibrosis, streptozotocin-induced type 1 diabetes, and fibrosis induced by a standard pellet diet with high fructose and UUO. The mechanism includes decreasing the proliferation of connective tissue collagen (Table 7); inhibiting oxidative stress by activating the Nrf2-HO-1/NQO1 pathway; inhibiting kidney fibrosis by suppressing TGF-β1/Smad3; increasing kidney ALKBH5 expression; reducing RNA m6A levels; restoring Klotho; and decreasing α-SMA expression. In vitro, genistein blocks kidney transdifferentiation and epithelial-to-mesenchymal transition (inhibits α-SMA and CTGF expression and restores E-cadherin expression) in human kidney tubular epithelial cells. Therefore, it is suggested that genistein has potential anti-kidney fibrosis effects, and further clinical research is needed.

### 4.7. The Effects of Genistein on Nephrotic Syndrome

Nephrotic syndrome is characterized by a series of symptoms, such as edema, proteinuria, hypoalbuminemia, and hyperlipidemia, and is common in children and adolescents [208]. However, when nephrotic syndrome is serious, it can cause harm to other organs of the body and even endanger life. Therefore, it is necessary to propose a new method for the treatment of nephrotic syndrome. A previous study [209] showed that genistein significantly increased the scores of pathological examinations by increasing the total antioxidant capacity and catalase activity and decreasing the contents of protein carbonyl and MDA in the kidneys of nephrotic model rats. In addition, the proliferation of the WEHI-164 kidney fibrosarcoma cell line in vitro was also inhibited by genistein in the study. Therefore, the result suggests that genistein may be used to reduce the symptoms of nephrotic syndrome in the future.

### 4.8. The Effects of Genistein on Menopausal Kidney Injury

Menopause is the period when a woman’s menstruation stops permanently, resulting in infertility. It usually occurs between the ages of 48 and 52. For those who have their uterus removed but their ovaries retained, hormone (e.g., estrogen) levels drop sharply, followed by menopause symptoms, usually including obesity; a metabolic imbalance in some tissues, such as the liver, bones, nervous system, and skeletal muscle [210]; and damage to the kidney tissue (e.g., glomerular and tubulointerstitial damage, even leading to diabetic nephropathy) [211]. Since genistein can exert estrogen-like effects, a previous study [212] found that genistein treatment (0.1% in the diet for 4 weeks) reduced insulin resistance and fat accumulation, promoted lipid metabolism, and improved abnormal protein expression induced by oxidative stress in ovariectomized rats fed a high-fat diet. Therefore, genistein can be considered to attenuate the kidney injury of climacteric women.

### 4.9. The Effects of Genistein on Aging-Induced Kidney Injury

Aging is a natural process that represents the cumulative changes in a person’s state over time, including a decline in physical functions, as well as disorders in psychological and social states, which eventually lead to the emergence of disease and death [213]. Previous studies have shown that genistein can improve aging-related diseases (e.g., bone loss [214] and Alzheimer’s disease [215]). According to a previous study [216], genistein treatment (2 and 4 mg/kg/day for 10 days) can also improve aging-induced kidney injury in male rats by decreasing age-related NF-κB activity (activated by angiotensin II during senescence) and the expression of downstream pro-inflammatory genes, which indicated that genistein had an obvious anti-inflammatory effect by inhibiting the activation of NF-κB induced by angiotensin II during aging. However, there is no relevant research on the effect of genistein on female aging (except for menopausal kidney injury), so it may be further discussed in the future.

## 5. The Mechanism of Genistein Actions in Kidney

Kidney diseases are very harmful to the human body, and there is no specific treatment, so it is necessary to find better functional components to delay the occurrence and development of kidney disease. As a phytoestrogen and isoflavone found in soybeans, genistein treatment can attenuate many kidney diseases (e.g., acute kidney injuries, kidney cancer, diabetic nephropathy, hypertensive kidney disease, kidney fibrosis, nephrotic syndrome, menopausal kidney damage, and aging-induced kidney injury).

The beneficial effects of genistein are mainly exerted through the following mechanisms: ① It reduces the inflammatory response by downregulating a variety of inflammatory factors (including IL-1, IL-6, NF-κB, TNF-α, TLR-4, and C-reactive protein). ② It inhibits oxidative stress (protein carbonyl, MDA, and Nrf2-HO-1/NQO1) that causes the oxidation of DNA, proteins, and lipids, thus affecting their structure and function. ③ Genistein can decrease arterial pressure through the inhibition of ACE, PKC-βII, NO/NOS, and vasoconstriction. ④ AGEs are formed through the non-enzymatic glycosylation of amino groups on proteins by reducing sugars or dicarbonyl [217], which is related to the pathogenesis of complications (e.g., kidney) of type 2 diabetes mellitus [218], and can be inhibited by genistein via the upregulation of glyoxalase I/II, AR, and FBG. ⑤ It reduces fibrosis by increasing ALKBH5 and Klotho and decreasing α-SMA and CTGF. ⑥ It inhibits apoptosis and promotes cell apoptosis through the inhibition of SIRT1 expression in common kidney diseases. ⑦ It inhibits proliferation and promotes apoptosis by depressing EED, HOTAIR/PRC2, SNAIL, and EGF and increasing CDKN2a in cancer. ⑧ It inhibits angiogenesis through the inhibition of VEGF and FGF in kidney cancer (Figure 4).

## 6. Conclusions

Genistein has many effects. For example, it can inhibit fibrosis; mesangial dilation; oxidative stress; and inflammatory cytokines, inhibiting mesangial dilation, inflammatory cytokines, oxidative stress, fibrosis, apoptosis, etc. In addition, genistein also has beneficial effects (prevents structural changes and antioxidation, improves activity, and reduces pathological damage) on a variety of cells in the kidney (e.g., mesangial cells, endothelial cells, and podocytes).

Genistein also plays an important role in the treatment of kidney disease. The effects include the modulation of renin release, calcium and phosphorus balance, excessive LPS, and kidney damage caused by a variety of acute and chronic diseases. In animal model experiments for many diseases, genistein reduced kidney damage caused by ischemia, reduced drug- and radiation-induced kidney injury, reduced kidney fibrosis, improved nephrotic syndrome, reduced postmenopausal kidney damage, reduced old-age-induced kidney damage, and reduced kidney barrier dysfunction by improving the glucose balance, reducing inflammation, increasing antioxidant activity and kidney function, reducing vascular resistance, and improving kidney blood pressure. Meanwhile, in cancer cells, genistein can prevent the abnormal proliferation of cancer cells, induce cancer cell apoptosis, and inhibit tumor growth. Genistein has a wide range of health benefits, which make it the first choice for the treatment of kidney disease. However, the effect of genistein on kidney cancer requires more studies in humans and animals.

In addition, there are a large number of cellular and animal experiments in this review, which may be used to predict whether genistein can prevent human kidney disease.

(1) However, at present, there is a lack of research exploring the relationship between the content and concentration of genistein in blood and kidney diseases. (2) Although many foods contain genistein (as mentioned earlier), the exact amount of genistein consumed in the same area and the concentration of genistein in the blood of different people may be different. (3) We can carry out more experiments using human pathological cell models to provide a more valuable reference for the application of genistein. Additionally, the following aspects need to be further studied and clarified: (i) the dose and bioavailability of genistein in different kidney diseases, (ii) genistein metabolism and biological effects, and (iii) the signaling mechanisms involved. (4) At present, only a limited number of studies have examined the role of genistein in human kidneys. Because there are no current ongoing clinical trials in humans, and the side effects and dosage of genistein in human kidneys, the most effective stages of kidney diseases, and the magnitude of the benefits are unknown. Therefore, more human trials are needed to address the above problems in the future.

## Figures and Tables

**Figure 1 nutrients-14-03768-f001:**
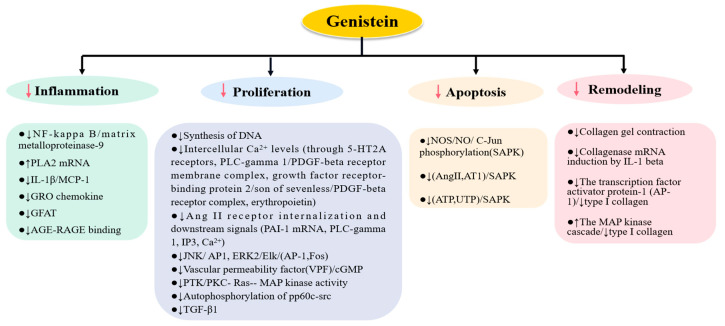
The effects of genistein on mesangial cells. NF-κB: nuclear factor NF-kappaB; PLA2: phospholipase A2 transcription; IL-1β: interleukin-1 beta; MCP-1: monocyte chemoattractant protein-1; GFAT: glutamine:fructose-6-phosphate amidotransferase; AGE: advanced glycation end products; RAGE: receptor for advanced glycation end products; Ca^2+^: calcium; 5-HT2A: 5-hydroxytryptamine 2A; PLC-γ1: phospholipase C-γ1; PDGF: platelet-derived growth factor; Ang II: angiotensin II; PAI-1: plasminogen activator inhibitor-1; IP3: inositol triphosphate; ERK2: extracellular signal-regulated kinase 2; AP-1: activating protein-1; Fos: c-fos gene; VPF: vascular permeability factor; cGMP: cyclic guanosine monophosphate; PTK: phototherapeutic keratectomy; PKC: protein kinase C; Ras: renin–angiotensin system; MAP: mitogen-activated protein; TGF-β1: transforming growth factor-β1; NO: nitric oxide; SAPK: c-Jun phosphorylation; ATP: adenosine-triphosphate; UTP: uridine triphosphate, ↓: inhibit; ↑: upregulate.

**Figure 2 nutrients-14-03768-f002:**
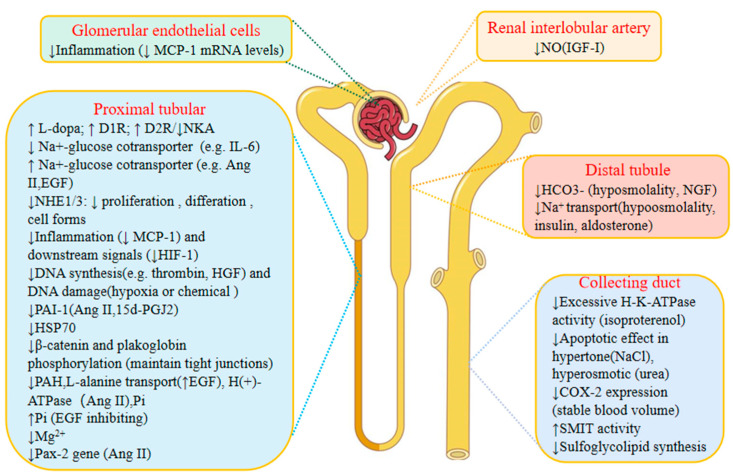
The effects of genistein on endothelial cells. MCP-1: monocyte chemoattractant protein-1; L-dopa: l-dihydroxyphenylalanine; DIR: D1 receptor; D2R: D1 receptor; EGF: epidermal growth factor; NHE: Na+/H+ exchanger; HIF-1: hypoxia-inducible factor 1; HGF: hepatocyte growth factor; PAH: phenylalanine hydroxylase; HSP70: heat shock protein 70; Pi: phosphate; Pax-2: paired homeobox-2 gene; COX-2: cyclooxygenase-2; SMIT: sodium/myo-inositol cotransporter, ↓: inhibit; ↑: upregulate.

**Figure 4 nutrients-14-03768-f004:**
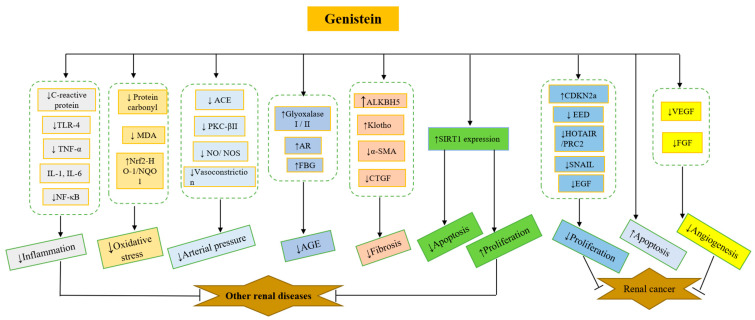
The mechanism of genistein actions in kidney. TLR-4: Toll-like-receptor-4, TNF-α: tumor necrosis factor α; IL-1β: interleukin-1 beta; IL-6: interleukin-6; NF-κB: nuclear factor NF-kappaB; MDA: malondialdehyde; Nrf2: nuclear factor erythroid 2-related factor 2; HO-1: heme oxygenase-1 (HO-1); NQO1: NAD(P) H:Quinone Oxidoreductase 1; ACE: angiotensin-converting enzyme; PKC: protein kinase C beta II; NO: nitric oxide; NOS: nitric oxide synthesis; AR: aldose reductase; FBG: fasting blood glucose; AGE: advanced glycation end products; CTGF: connective tissue growth factor; CDKN2a: cyclin-dependent kinase inhibitor 2a; EED: embryonic ectoderm development; HOTAIR: HOX transcript antisense RNA; PRC2: polycomb repressive complex 2; EGF: epidermal growth factor; VEGF: vascular endothelial growth factor; FGF: fibroblast growth factor, ↓: inhibit; ↑: upregulate.

**Table 2 nutrients-14-03768-t002:** The effects of genistein on podocytes in animal models.

Cells	Treatments	Effects and Mechanisms	Ref.
Mouse podocyte cells	20 μM, 30 min prior to treatment for 24 h	Decreasing D-ribose-induced ceramide accumulation, EV release and IL-1β secretion, and NLRP3 inflammasome	[105]
Rat podocytes	200 µM for 4 d	Decreasing the expression of a-SMA protein and the percentage of a-SMA-positive cells stimulated by TGF-β_1_	[106]
Mouse podocyte cell line, H-2K^b^-tsA58, with high D-glucose	20 µM for 6 h	Maintaining the level of autophagy by inactivating mTOR signaling and the level of MyD88 siRNA	[107]
Rat primary podocytes	200 µM for 4 h	Causing apoptosis of podocytes	[108]
Mouse podocyte cell line	60 μM for 20 h	Increasing cell loss under fluid flow stress	[109]

**Table 3 nutrients-14-03768-t003:** The effects of genistein attenuating kidney cancer cells.

Kidney Cancer Cell Lines	Treatments (Genistein)	Effects and Mechanisms	Ref.
SMKT-R3 (human)	50 g/mL for 15 min	Inhibiting tyrosine kinases and glycolipid sulfotransferase	[152]
GRC-1 (human)	20 and 40 mM/L for 72 h	Inhibiting the proliferation of kidney cell carcinoma cells; causing cell cycle arrest at the G_1_/M and G_2_/S phases	[153]
kidney carcinoma cells SMKT-R-1,3 (human)	4, 40, and 100 μg/mL under hypoxic conditions for 12 h	Suppressing the expression of the angiogenic factors vascular endothelial growth factor and basic FGF	[154]
A498, ACHN, and HEK-293 (human)	10, 25, and 50 μ mol/L for 3 d	Inhibiting proliferation by decreasing DNA Methyltransferase and methyl-CpG-binding domain 2 activity and increasing HAT activity and induction of cell cycle arrest	[155]
SMKT R-1, 2, 3, 4 lines (human)	50 and 100 mg/mL for 48 h	Inhibiting cell proliferation, inducing apoptosis, and suppressing in vivo angiogenesis	[156]
A-498; ATCC numbers: HTB44, HTB-47, 786-O, CRL-1932, and Caki-2 (human)	25 µM for 4 d	Inhibiting Wnt signaling by regulating miR-1260b expression	[157]
Human clear cell kidney carcinoma cell lines (ccRCC) (human)	25 µM for 96 h	Reducing cell proliferation and migration by suppressing EED levels in PRC2 HOTAIR/PRC2 interaction, HOTAIR /PRC2 recruitment to the ZO-1 promoter, and enhancing ZO-1 transcription; inhibiting SNAIL transcription by reducing HOTAIR/SMARCB1 interaction	[158]
HEK293, HK-2, 786-O, CAKI-1, 769-P, and CAKI-2 cell lines (human)	25, 50, and 100 µM for 5 d	Inducing cell apoptosis and inhibiting cell proliferation of kidney cancer cells by increasing the expression of CDKN2a and decreasing CDKN2a methylation	[159]
A-498 cells in nude mice (mouse)	25 µM for 4 d	Inhibiting the expression of miR-21 in A-498 cells and in the tumors	[160]
Kidney carcinoma cell (mouse)	0.2 mL, 80 mg/kg/day, injected once a day for 14 d	Suppressing tumor growth and decreasing MVD and VEGF levels	[161]

**Table 5 nutrients-14-03768-t005:** The effects of genistein attenuating hypertensive kidney disease.

Animal	Model	Treatments (Genistein)	Effects and Mechanisms	Ref.
Wistar rats	Fructose-fed hypertensive	1 mg/kg/day in diet for 60 d	Lowering BP by restoring ACE, PKC-βII, and eNOS expression and preserving kidney ultrastructural integrity	[181]
Sprague-Dawley rats	2-kidney 1-clip kidney hypertensive	5.0 mg/kg/day for 8 weeks	Restoring nitric oxide, NOS activity, phosphorylated eNOS expression, and cGMP	[182]
SHR-SPs	Dietary NaCl with hypertension	0.6 mg/g diet for 9 weeks	Blunting a dose-related increase in arterial pressure	[183]
Wistar rats	Isolated perfused rat kidney	15 mg/kg for 24 h	Reducing kidney vascular resistance relative to vehicle in isolated perfused kidney	[115]

**Table 7 nutrients-14-03768-t007:** The effects of genistein attenuating kidney fibrosis.

Animal	Model	Treatments (Genistein)	Effects and Mechanisms	Ref.
C57BL mice	UUO-induced kidney interstitial fibrosis	10 mg/kg/body weight i.p. 24 h prior to the UUO for 7 d	Increasing kidney ALKBH5 expression, reducing RNA m6A levels, and ameliorating kidney damage.	[203]
Sprague-Dawley rats	Streptozotocin-induced diabetic	5 and 25 mg/kg, daily gavage for 8 weeks	Inhibiting oxidative stress by activating the Nrf2-HO-1/NQO1 pathway and alleviating kidney fibrosis by inhibiting the TGF-β1/Smad3 pathway	[204]
C57BL/6 mice	Kidney fibrosis, UUO-induced	10 mg/kg, intraperitoneal injection daily administered 1 day before UUO	Restoring Klotho via epigenetic histone acetylation and DNA demethylation	[205]
Wistar rats	Standard pelletdiet (fructose-fed)	1 mg/kg/day for 45 d	Decreasing α-SMA expression and mitigating proliferation of connective tissue collagen deposition in perivascular and intraglomerular regions	[206]
Human kidney tubular epithelial HK-2 cells	PTH-induced kidney interstitial fibrosis	1, 25, 50, and 100 µM for 30 min	Inhibiting PTH-induced α-SMA expression, restoring E-cadherin expression, decreasing mRNA, protein expression, and activity of CTGF	[207]

## Data Availability

Not applicable.

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
