# Peer review of "Effects of Genistein on Common Kidney Diseases"

_nutrients, 2022, doi:10.3390/nu14183768_

Round 1

Reviewer 1 Report

Thank you for the opportunity to peer review the manuscript, “Effects of Genistein on Common Kidney Diseases”

Abstract Lines 24-25: The paper reviews the animal and human studies on the protective effects of genistein on kidney in vivo and in vitro so as to provide reference for clinical research in the future.

The strengths of the manuscript are the illustrations and exhaustive review of the literature.

After careful review, my recommendation is for rejection or major revision based on the following:

1.       The manuscript needs extensive editing for English language. The authors should be recognized for their initial efforts in this area but more assistance is needed particularly in verb tenses and sentence structure.

2.       The majority of the data is from animal studies.  The type of study is often lost in the description.  With so few human studies, I might suggest restricting to animal research and adding a separate paragraph and table highlighting the few human studies.

3.       Authors might consider a stronger introduction on genistein rather than kidney physiology. The introduction should end with a clear purpose statement that aligns with the abstract.

4.       The use of “kidney” is the preferred term rather than renal.

5.       Authors need to carefully review claims, many of which are not supported in the literature or the authors’ cite a single study.

a.       Example: Lines 51+ using citations 2-4. The human literature does not cite the use of these nutrients to effect kidney disease incidence or progression. Animal literature not investigated from this claim.

b.       Example: Line 133 – Statement claims genistein can block development of renal disease. Statement not supported by literature.

c.       Example: Line 374, ref 150.  Renal cancer is one of top 10 killers.  Concern citation not correct or needs to be grounded in the population where this statistic occurs.

6.       Authors are suggested to update older citations for latest citations.  Example: Ref 180 for hypertension is from 1980.

7.       Genistein is the focus but very little information is given on the practical application or understanding. 

a.       Consider adding a table illustrating the amount of genistein in 100 g of food items Can the amount of genistein studied be incorporated as a nutrient through diet OR does it have to be a purified nutrient administered as a dietary supplememt?

b.       How was genistein modified/purified in the cited examples? The authors do not integrate potential dose or type of genistein used in the studies.

c.       Line 432 cites dosage for a rat study was 1 mg/kg/d.  How does this compare to other studies and is dose variable for focus (MC cells for example versus another target)

Author Response

Dear professor:

Thank you very much for your advice. We have carefully revised the manuscript "Effects of Genistein on Common Kidney Diseases " (number 1852280) submitted to Nutrients, point by point, according to your comments and re-submitted it for your consideration.Please see the attachment for the revised manuscript.

Sincerely,

All authors

Point 1: The manuscript needs extensive editing for English language. The authors should be recognized for their initial efforts in this area but more assistance is needed particularly in verb tenses and sentence structure

Response 1: Thank you for your valuable advice. We have used institutions recommended by journals to help us revise the language of the manuscript in the hope that it can meet the requirements of the publishing house.

Point 2:The majority of the data is from animal studies.  The type of study is often lost in the description.  With so few human studies, I might suggest restricting to animal research and adding a separate paragraph and table highlighting the few human studies.

Response2: Thank you for your valuable comments. Because there are few human experiments, we have stated them separately in kidney diseases.

Point 3: Authors might consider a stronger introduction on genistein rather than kidney physiology. The introduction should end with a clear purpose statement that aligns with the abstract. 

Response 3:Thank you for your valuable advice. We have adjusted the primary and secondary relationship between genistein and kidney, and the introduction is consistent with the abstract.

Point 4: The use of “kidney” is the preferred term rather than renal

Response 4: Thank you for your valuable comments. We have changed all “renal” into “kidney“in the full text.

Point 5: Authors need to carefully review claims, many of which are not supported in the literature or the authors’ cite a single study.

  1. Example: Lines 51+ using citations 2-4. The human literature does not cite the use of these nutrients to effect kidney disease incidence or progression. Animal literature not investigated from this claim.

Response a: Thank you for your valuable comments. Due to the lack of support in the literature, the statement that nutrients affect kidney disease has been deleted.

  1. Example: Line 133 – Statement claims genistein can block development of renal disease. Statement not supported by literature.

Response b: Thank you for your valuable comments. In our version, line 133 shows that genistein promotes collagen synthesis mainly by inhibiting tyrosine kinase and collagen degradation. The genistein obtained below can improve the function of cells and slow down the development of kidney diseases

  1. Example: Line 374, ref 150.  Renal cancer is one of top 10 killers.  Concern citation not correct or needs to be grounded in the population where this statistic occurs.

Response c: Thank you for your valuable comments. For the sake of the rigor of the article, we deleted the statement that renal cell carcinoma is one of the top ten killers.

Point 6:  Authors are suggested to update older citations for latest citations.  Example: Ref 180 for hypertension is from 1980

Response 6: Thank you for your valuable comments. We have updated the latest citations.

Point 7:  Genistein is the focus but very little information is given on the practical application or understanding.

  1.       Consider adding a table illustrating the amount of genistein in 100 g of food items Can the amount of genistein studied be incorporated as a nutrient through diet OR does it have to be a purified nutrient administered as a dietary supplememt?

Response a: Thank you for your valuable advice. The content of genistein in various foods is added to Table 1 of the introduction. In addition, the content of genistein in food may meet the needs of the human body, but because 1) the bioavailability of genistein in food is less than that of genistein intake, 2) individuals have different choices of food types, and they may not like to eat large amounts of foods with high levels of genistein,it may be more suitable to use purified genistein.

Table 1. The content of genistein in various foods

Foods

content (μg / 100g dry weight)

Soybean

26800-102500

Kidney bean

18.0-518.0

Chickpea

69.0-214.0

Pea

0-49.7

Lentil

7.0-19.0

Kudzu leaf

2520

Kudzu root

12600

Black gram

1900

Alfalfa

5.0

Peanut

8.0

Caraway seed

64.0

Sunflower seed

13.9

Barley

7.7

Broccoli

8.0

Cauliflower

9.0

  1.       How was genistein modified/purified in the cited examples? The authors do not integrate potential dose or type of genistein used in the studies.

Response b: Thank you for your valuable advice. The commonly used purification methods have been added to the introduction of genistein in the text, and the commonly used types are pure. In addition, we checked the article and added a description of genistein dosage in the experiment.

  1.       Line 432 cites dosage for a rat study was 1 mg/kg/d.  How does this compare to other studies and is dose variable for focus (MC cells for example versus another target)

Response c: Thank you for your valuable advice.  In different states of kidney disease, the content of genistein is not completely consistent. For example, in insulin resistance induced by a high-fructose model, the damage to the kidney is relatively small; therefore, the dose of genistein should be adjusted according to different kidney diseases and different pathological conditions. However, according to the current literature survey, there is no unified standard, so the dose of genistein requires further clinical research.

Reviewer 2 Report

Interesting, original and well-structured work that reports on the effects and usefulness of genistein in kidney disease, I think it would be convenient, although much remains to be tested, to identify the possible uses and in what type of patients with kidney disease more concisely. Possible beneficial effects on blood pressure control are explained, it could be considered in the future as a coadjuvant treatment for its control. Review the discussion so that, although it has not been definitively proven, its possible usefulness in renal patients is more concrete.

Interesting, original and well-structured work that reports on the effects and usefulness of genistein in kidney disease,

  I think it would be convenient, although much remains to be tested, to identify the possible uses and in what type of patients with kidney disease more concisely. Possible beneficial effects on blood pressure control are explained, it could be considered in the future as a coadjuvant treatment for its control. Review the discussion so that, although it has not been definitively proven, its possible usefulness in renal patients is more concrete.

Author Response

Dear professor:

Thank you very much for your advice.

Please see attachment for revised manuscript

Sincerely,

All authors

Point : Interesting, original and well-structured work that reports on the effects and usefulness of genistein in kidney disease, I think it would be convenient, although much remains to be tested, to identify the possible uses and in what type of patients with kidney disease more concisely. Possible beneficial effects on blood pressure control are explained, it could be considered in the future as a coadjuvant treatment for its control. Review the discussion so that, although it has not been definitively proven, its possible usefulness in renal patients is more concrete.

Response: Thank you for your approval of our article. We will revise this article and strive to make it more suitable for publication.

Reviewer 3 Report

Authors wrote an extensive review regarding genistein on kidney diseases. 

Although the current manuscript provided extensive knowledge on genistein but are too lengthy and includes irrelevant or self evident contents.

* Although the review should be extensive but it should not be too lengthy but be concise and to the point. Authors should tried to shorten the manuscript to two-thirds to three-fourth.

* The 1st paragraph of the Introduction is too basic for the readers of Nutrients, especially who are interested in kidney diseases, so should be omitted.

* Most of the references are on experiments using cells or animals. Authors should discuss how it is valid to apply these cell/animal results to human who has very complex and redundant cell signaling systems.

* Is there any food product which contains phytoestrogens that is available and we can eat? Is there any evidence of those product which makes us understand the beneficial (or detrimental) effects on kidney or human.

* English are extensively checked by native speaker.

Author Response

Response to Reviewer Comments

Dear professor:

Thank you very much for your advice. We have carefully revised the manuscript "Effects of Genistein on Common Kidney Diseases " (number 1852280) submitted to Nutrients, point by point, according to your comments and re-submitted it for your consideration.Please see attachment for revised manuscript

Sincerely,

All authors

Point 1: Although the review should be extensive but it should not be too lengthy but be concise and to the point. Authors should tried to shorten the manuscript to two-thirds to three-fourth.

Response 1: Thank you for your valuable comments. We have shortened the manuscript appropriately as suggested.

Point 2: The 1st paragraph of the Introduction is too basic for the readers of Nutrients, especially who are interested in kidney diseases, so should be omitted.

Response 2: Thank you for your valuable comments. Based on your suggestions, we have shortened the first paragraph and merged it with the second paragraph, and deleted reference 1.

Point 3:  Most of the references are on experiments using cells or animals. Authors should discuss how it is valid to apply these cell/animal results to human who has very complex and redundant cell signaling systems.

Response 3:Thank you for your valuable comments. We have added statements in the third paragraph and at the beginning of the fourth paragraph of the conclusion: “In addition, there are a large number of cellular and animal experiments in this review, which may be used to predict whether genistein can prevent human kidney disease .However, at present,there is a lack of research to explore the relationship between the content and concentration of genistein in blood and kidney diseases.”

Point 4: Is there any food product which contains phytoestrogens that is available and we can eat? Is there any evidence of those product which makes us understand the beneficial (or detrimental) effects on kidney or human.

Response 4:Thank you for your valuable comments. We have added statements in the middle of the fourth paragraph of the conclusion:“Although many foods contain genistein (as mentioned earlier), the exact amount of genistein consumed in the same area and the concentration of genistein in the blood of different people may be different. We can do more experiments with human pathological cell model to provide more valuable reference for the application of genistein”

Point 5: English are extensively checked by native speaker.

Response 5:Thank you for your valuable advice. We have used institutions recommended by journals to help us revise the language of the manuscript in the hope that it can meet the requirements of the publishing house.

Round 2

Reviewer 1 Report

The authors have made extensive revisions to the manuscript.  They have responded to all the reviewer questions.  I am still slightly concerned about some of the inferences of animal models to human clinical practice, but I realize research is ongoing,  I would suggest review of the summary statement as to my knowledge, there are no current ongoing clinical trials in humans so risks=benefits, side-effects, effect of doses, stage of kidney disease when most effective - etc.  These are all unknown and should be in the summary to promote next steps

Table 1: cite where data coming from; some have wide ranges.  Use of "dried" as a term may be misleading as soybean is usually as tofu or hydrated portion?  A footnote may be helpful to explain'
Table 2.  Add "in animal models" to the title

Author Response

Dear professor:

Thank you very much for your advice. We have carefully revised the manuscript. Please see the attachment.

Sincerely,

All authors

Reviewer 3 Report

Revised appropriately. No further suggestions/comments.

Author Response

Dear professor:

     Thank you for your approval of our article.

Sincerely,

All authors